# Mutational scanning reveals the determinants of protein insertion and association energetics in the plasma membrane

Assaf Elazar, Jonathan Weinstein, Ido Biran, Yearit Fridman, Eitan Bibi, Sarel Jacob Fleishman*

Department of Biomolecular Sciences, Weizmann Institute of Science, Rehovot, Israel

**Abstract** Insertion of helix-forming segments into the membrane and their association determines the structure, function, and expression levels of all plasma membrane proteins. However, systematic and reliable quantification of membrane-protein energetics has been challenging. We developed a deep mutational scanning method to monitor the effects of hundreds of point mutations on helix insertion and self-association within the bacterial inner membrane. The assay quantifies insertion energetics for all natural amino acids at 27 positions across the membrane, revealing that the hydrophobicity of biological membranes is significantly higher than appreciated. We further quantitate the contributions to membrane-protein insertion from positively charged residues at the cytoplasm-membrane interface and reveal large and unanticipated differences among these residues. Finally, we derive comprehensive mutational landscapes in the membrane domains of Glycophorin A and the ErbB2 oncogene, and find that insertion and self-association are strongly coupled in receptor homodimers.

*For correspondence: sarel@weizmann.ac.il

Competing interests: The authors declare that no competing interests exist.

## Introduction

The past four decades have seen persistent efforts to decipher the contributions to membrane-protein energetics (*Reynolds et al., 1974*; *Cymer et al., 2015*). Membrane-protein folding can be conceptually divided into two thermodynamic stages (*Popot and Engelman, 1990*; *Cymer et al., 2015*), each of which affects membrane-protein structure, function, and expression levels: the insertion into the membrane of transmembrane segments as α helices, and their association to form helix bundles (*Ben-Tal et al., 1996*; *Heinrich and Rapoport, 2003*; *Moll and Thompson, 1994*; *White and Wimley, 1999*; *Popot and Engelman, 1990*). While significant progress has been made in structure prediction, design, and engineering of soluble proteins (*Fleishman and Baker, 2012*), important but fewer successes were reported in design of membrane proteins (*Joh et al., 2014*; *Li et al., 2004*), largely owing to the complexity of the plasma membrane and the lack of systematic and accurate measurements of membrane-protein energetics (*Cymer et al., 2015*).

Recently, experimental systems that offer a realistic model for biological membranes have advanced. von Heijne and co-workers quantitated the partitioning of engineered peptides fused to the bacterial transmembrane protein, leader peptidase (Lep), between membrane-inserted and translocated states, and highlighted the importance of interactions between the translocon and the nascent polypeptide chain in determining partitioning (*Hessa et al., 2007*; *Öjemalm et al., 2013*). The insertion energetics obtained from this assay, however, were significantly lower than expected from previous theoretical and experimental studies; for instance, the apparent atomic-solvation

**eLife digest** Cells are defined by a thin membrane that separates the inside of the cell from the outside. The core of this membrane is hydrophobic, meaning that it repels water. Many signals and nutrients cannot pass through the membrane itself, but can pass through the proteins that span the membrane. Membrane proteins are therefore essential for living cells; yet even after decades of research, it remains unclear how proteins interact with the membrane and which features determine a protein's stability in a biological membrane.

Since the early 1980s it was known that the bacterium *E. coli* could grow on a common antibiotic called ampicillin if it had enough of an antibiotic-degrading enzyme called β-lactamase anchored into its inner membrane. Now, Elazar et al. have used this enzyme to obtain detailed information on the interactions between a biological membrane and a membrane protein. First, hundreds of different mutations were introduced into the gene that encodes the enzyme to generate a population of bacteria that each had a slightly different membrane anchor. The mutant bacteria were then grown in the presence of the antibiotic, meaning that those mutants with a more stable membrane anchor were more likely to survive and grow than those with less stable anchors.

Elazar et al. then collected all the surviving bacteria, sequenced their DNA and measured how common the different mutations were in the final population. This approach was less labor-intensive and more accurate than traditional methods for monitoring membrane-anchored proteins, and the resulting large dataset was used to uncover which features affect a protein's stability in a membrane. These results also showed that a biological membrane's core is considerably more hydrophobic than was previously thought.

In addition to being hydrophobic, biological membranes have more negative charge in the side that faces into the cell. This means that membrane proteins with a positive charge in this region will be more stable, and Elazar et al. were able to use their new system to measure this effect for the first time.

Finally, membrane proteins do not only span the membrane; they also bind with other membrane proteins in order to carry out their roles. Elazar et al. used their system to look at the surfaces of human membrane proteins that interact with one another, and build a detailed map of the interaction surfaces, from which they derived accurate models of the membrane proteins.

Overall, these new findings could now be used to model the three-dimensional structures of membrane proteins and improve their stability. This in turn may help efforts to develop these proteins into more robust experimental tools and in the search for drugs that target membrane proteins.

parameter, which quantifies the free-energy contribution from the partitioning of hydrophobic surfaces to the membrane core, was only 10 cal/mol/$Å^2$ according to the Lep measurements (*Ojemalm et al., 2011*), compared to ~30 cal/mol/$Å^2$ from previous analyses (*Andrew Karplus, 1997*; *Vajda et al., 1995*). Additionally, the magnitude of the insertion free energies for individual amino acids were substantially lower according to the Lep system (*Hessa et al., 2007*; *Ojemalm et al., 2011*; *Öjemalm et al., 2013*) compared to other studies (*Moon and Fleming, 2011*; *Shental-Bechor et al., 2006*). These discrepancies led to suggestions that the Lep measurements were 'compressed' relative to others due to interactions between the engineered protein and other membrane constituents (*Johansson and Lindahl, 2009*; *Shental-Bechor et al., 2006*).

Membrane-protein energetics are governed not only by the insertion but also by the association of helices into bundles. A significant body of work has shown that association is driven by packing interactions and short sequence motifs comprising small-xxx-small residues, where small is any of the small polar residues (Ser, Gly, or Ala) and x is any residue (*Russ and Engelman, 2000*; *Senes et al., 2004*; *Melnyk et al., 2004*). However, while it is recognized that insertion and association both play roles in protein energetics (*Duong et al., 2007*; *Finger et al., 2006*; *Moll and Thompson, 1994*; *Ben-Tal et al., 1996*; *Heinrich and Rapoport, 2003*; *Popot and Engelman, 1990*), the interplay between these two aspects has not been subjected to systematic experimental analysis. Given the

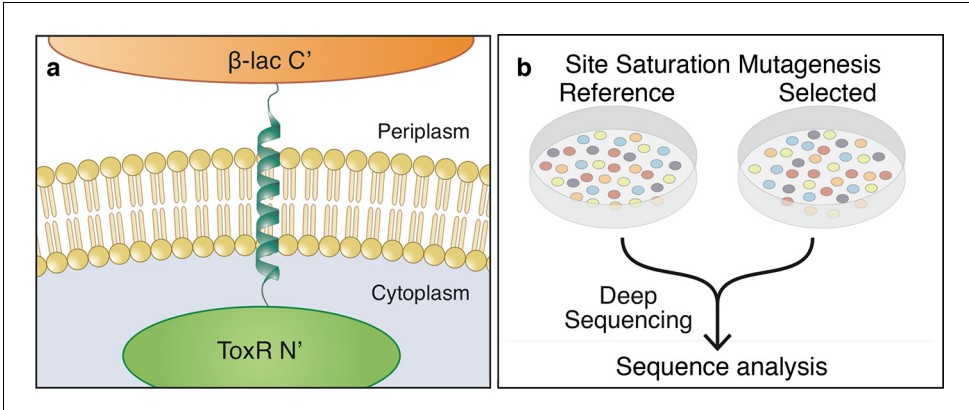

**Figure 1.** The dsTβL assay for measuring the sequence determinants of membrane-protein energetics. (a) The TβL genetic construct fuses a membrane segment to two antibiotic selection markers: β-lactamase and ToxR, which report on insertion and self-association, respectively. (b) Libraries encoding every point mutation of a membrane segment are plated on selective and non-selective medium. Following overnight growth, the libraries are extracted and DNA segments, which encode the membrane domain, are subjected to deep-sequencing analysis.

The following source data is available for figure 1:

**Source data 1.** Deep-sequencing read quality is high throughout the membrane-spanning segment.

**Source data 2.** Deep-sequencing counts for each mutant.

remaining open questions on membrane-protein and protein-protein interactions within the membrane, we established a high-throughput assay to shed light on both factors and their coupling in a systematic and unbiased way within the bacterial plasma membrane.

## Results

### dsTβL: a high-throughput assay for measuring membrane-protein energetics

To overcome gaps in our understanding of membrane-protein energetics, we adapted the TOXCAT-β−lactamase (TβL) screen (*Lis and Blumenthal, 2006*; *Russ and Engelman, 1999*; *Langosch et al., 1996*) for high-throughput analysis by deep mutational scanning (*Boucher et al., 2014*; *Fowler and Fields, 2014*); we refer to this new method as deep-sequencing TOXCAT-β−lactamase (dsTβL). TβL is a genetic screen based on a chimera, in which a membrane-spanning segment is flanked on the amino terminus by the ToxR dimerization-dependent transcriptional activator of a chloramphenicol-resistance gene and on the carboxy terminus by β-lactamase (*Figure 1a*). In this construct, bacterial survival on ampicillin monitors membrane integration (*Broome-Smith et al., 1990*; *Jaurin et al., 1982*), and survival on chloramphenicol correlates with self-association of the membrane span (*Lis and Blumenthal, 2006*; *Russ and Engelman, 1999*; *Langosch et al., 1996*). Furthermore, β-lactamase and ToxR function only in the periplasm and cytoplasm, respectively; therefore, unlike previous assays of membrane-protein insertion (*Hessa et al., 2007*), the orientation of the inserted segment relative to the membrane is known, and only proteins inserted with their carboxy terminus located in the periplasm would be selected. Most studies using the TOXCAT screen fused maltose-binding protein (MBP) in the carboxy-terminal domain (instead of β-lactamase) and used the MBP-null *E. coli* strain MM39 and maltose as sole carbon source to monitor membrane integration (*Russ and Engelman, 1999*; *Langosch et al., 1996*; *Melnyk et al., 2004*; *Li et al., 2004*). Unlike this conventional TOXCAT experiment, in TβL survival on ampicillin depends linearly on β−lactamase expression levels (*Broome-Smith et al., 1990*; *Jaurin et al., 1982*; *Li et al., 2004*; *Lis and Blumenthal, 2006*), thus providing a more sensitive reporter for membrane insertion than MBP. Furthermore, the MM39 strain is not amenable to high-throughput transformation as required

by our study; with TβL, we were able to use the high-transformation efficiency E. cloni cells (see Materials and methods).

Previous studies based on ToxR activity measured the effects of mutations using colony growth and enzyme-linked immunosorbant assay (ELISA), which do not allow high-throughput analysis (*Mendrola et al., 2002*; *Langosch et al., 1996*; *Lis and Blumenthal, 2006*; *Russ and Engelman, 2000*; *Melnyk et al., 2004*). Here, instead, we subject libraries encoding every amino acid substitution in the membrane domain to selection on agar plates containing either ampicillin alone or ampicillin and chloramphenicol to monitor insertion and self-association, respectively (*Figure 1b*); the same bacterial population is also plated on non-selective agar and serves as a reference to control for clonal-representation biases. Following overnight growth, the bacteria in each plate are pooled, plasmids encoding the TβL construct are extracted from each pool, and the variable gene segment, which encodes the membrane span, is amplified by PCR. The three resulting DNA samples are subjected to deep sequencing, which reports the relative frequency of each mutant in the selected and reference populations (*Boucher et al., 2014*) (see *equation 1* in Materials and methods). If the cytoplasmic protein fraction were perfectly constant among different mutants, the measured population frequencies could be interpreted as the relative propensies of each mutant to insert into the membrane or to self-associate in the membrane. This condition is unlikely to hold for all mutants; still, the agreement reported below with multiple lines of biophysical evidence on purified proteins suggests that the population frequencies provide a reasonable measure for changes in membrane-insertion and self-association partitioning. Hence, if we treat the population frequencies as if the mutants' partitioning between cytosolic, membrane-inserted, and self-associated fractions were under thermodynamic control, following the Boltzmann equation we can derive, at each position $i$ in the membrane span, apparent free energy changes for insertion or self-association due to the substitution from wild type to amino acid $aa$, $\Delta\Delta G_{aa,i}^{app}$ (see *equations 2–3* in Materials and methods). Although confounding factors, such as nonspecific interactions between the inserted segment and other bacterial membrane proteins, may affect the readout from the experiment, insertion and self-association are likely to dominate, since every mutant in this library differs from the wild type by only one amino acid; furthermore, all mutants are subjected to identical selection conditions, including temperature and antibiotic, thereby minimizing experimental noise (*Mackenzie and Fleming, 2008*).

## Systematic per-position contributions to membrane-protein insertion

We used dsTβL to comprehensively map the sequence determinants of membrane insertion in a single-pass membrane segment (*Figure 2a*). To minimize the effects of self-association on experimental readout, we chose the C-terminal portion of human L-Selectin (CLS), which does not self-associate (*Srinivasan et al., 2011*) (*Figure 2—figure supplement 1*). Furthermore, the CLS amino acid sequence includes several polar amino acids (*Figure 2a*, bottom); we therefore reasoned that its membrane-expression levels would be sensitive even to point mutations. To verify that the deep-sequencing data reflected trends observed in experiments with single clones, we selected 10 single-point substitutions at the membrane-spanning segment's amino terminus and at its core, and subjected them to experimental analysis on a clone-by-clone basis. Each clone and the wild type were grown overnight in non-selective medium, normalized to the same density, and plated in serial dilutions on ampicillin-containing agar to estimate relative viability (*Figure 2—figure supplement 2*). Nine of the 10 selected single-point substitutions (all but Val302Lys) showed qualitatively similar trends of viability in deep sequencing and single-clone analysis. The resolution of the deep-sequencing data, however, is much greater than that seen in the single-clone assays; for instance, whereas both charge mutations, Met303Glu and Ala311Arg, are highly deleterious according to deep sequencing and plate viability, the $\Delta\Delta G_{insertion}^{app}$ value from deep sequencing for the former is 3.7 kcal/mol compared to 1.3 kcal/mol for the latter, emphasizing the larger dynamic range of deep sequencing compared to traditional viability screens. We next expressed these 10 mutants in non-selective conditions, isolated membrane preparations for each (*Molloy, 2008*), and measured membrane-localization levels relative to wild type using Western blots with an antibody targeting β-lactamase (*Figure 2—figure supplement 3* see Materials and methods). All clones expressed in the membrane fraction and ran at the expected size of ~55 kDa. Of the 10 tested mutations, 6 showed the expected trends, including mutations that increased (Met303Leu, Val304Phe, and Ala311Leu) or

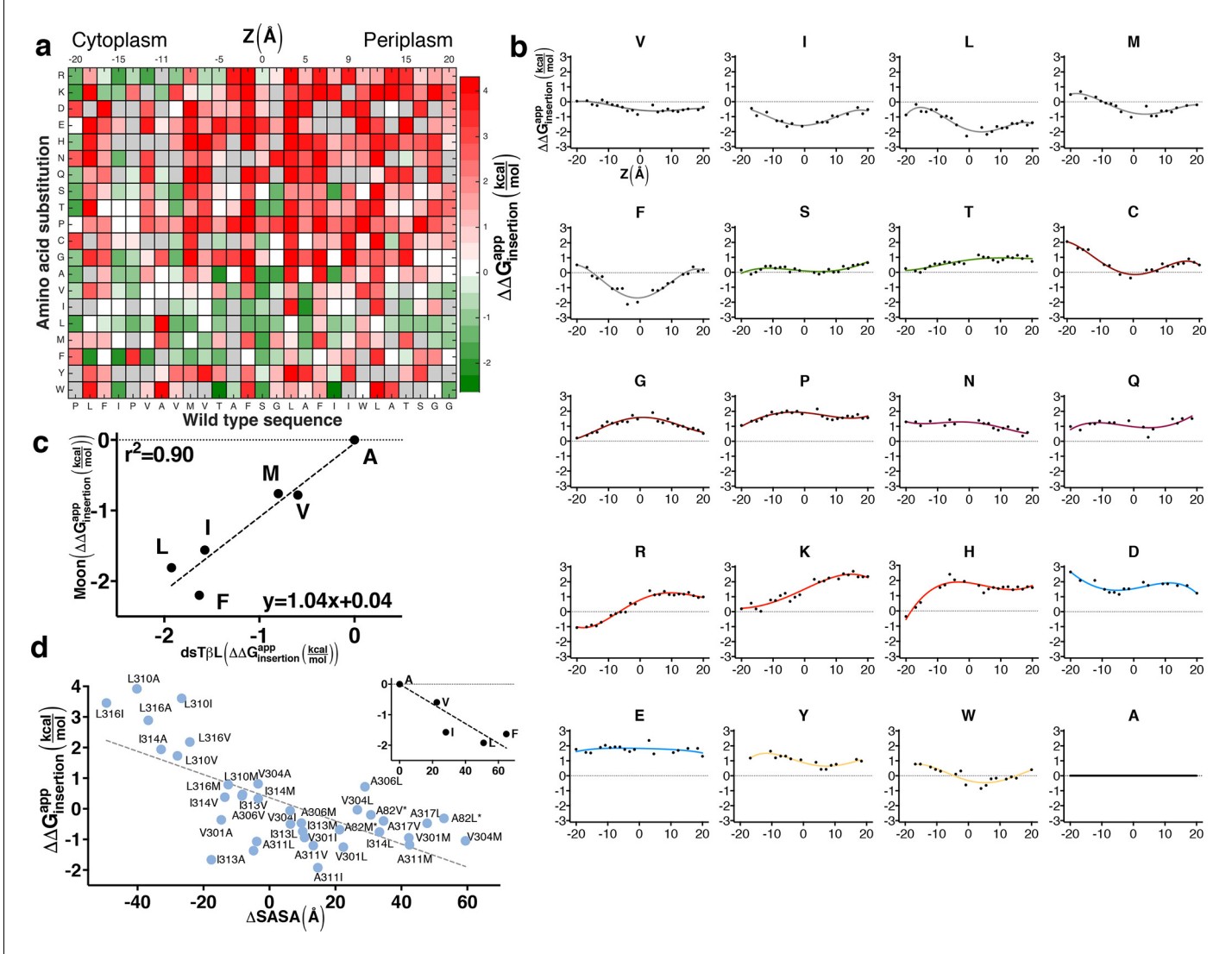

**Figure 2.** The sequence determinants of protein expression in native bacterial membranes. (a) Each tile reports the apparent change in free energy $\Delta\Delta G_{insertion}^{app}$ relative to wild type for every CLS point mutant (see *equation 3* in Materials and methods). Gray tiles mark substitutions that were eliminated from the analysis due to low counts (<100) in the reference population. (b) Per-position insertion profiles for each amino acid residue. (c) Comparison of dsTβL insertion results at the plasma membrane mid-plane (Z = 0) with values from the Moon scale (*Moon and Fleming, 2011*). (d) The apparent atomic-solvation parameter is the slope of the linear regression of $\Delta\Delta G_{insertion}^{app}$ and computed change in solvent-accessible surface area (SASA) due to each mutation (slope = -37 cal/mol/Å², $r^2$ = 0.48, p<0.0001) (see Materials and methods). (*inset*) Inferring the atomic-solvation parameter from the relationship of $\Delta\Delta G_{insertion}^{app}$ at Z = 0 for aliphatic residues and their change in SASA computed on a model poly-Ala α helix (slope=-32 cal/mol/Å², p = 0.002). CLS, C-terminal portion of human L-Selectin

The following figure supplements are available for figure 2:

**Figure supplement 1.** Human CLS and its single-site mutation library do not self associate.

**Figure supplement 2.** Deep-sequencing data reflects trends observed in clone-by-clone experiments.

**Figure supplement 3.** Western blot quantification of membrane expression of TβL mutants.

**Figure supplement 4.** Comparison of dsTβL insertion results at the membrane mid-plane with published hydrophobicity scales.

decreased expression (Val302Lys, Leu310Ala, and Ala311Arg) in agreement with the deep-sequencing and single-clone viability data (*Figure 2—figure supplement 3*). For instance, Ala311Arg was much less viable and showed lower membrane localization than wild type, whereas Ala311Leu was more viable and had higher membrane localization. However, three mutants to charges at the amino terminus (Val311His, Met303Glu, and Val304Asp) increased expression levels according to Western blots but were disruptive according to dsTβL. We attribute this difference to the fact that ampicillin viability reports not only on membrane-expression levels but also on the appropriate membrane integration, which is not captured by Western blots.

We next computed the apparent free-energy change of each substitution across the membrane relative to a substitution to Ala, and at each position $i$ computed the running average over five neighboring positions [$i$-2...$i$+2] (*Figure 2b*; *Supplementary file 1*). The resulting profiles describe the energetics of inserting each of the twenty amino acids relative to Ala at each position across the bacterial plasma membrane (*Figure 2b*). Although the location of the membrane mid-plane ($Z = 0$) could not be determined unambiguously in this assay, we estimated it by aligning the hydrophobic residues' profiles (Leu, Ile, Met, and Phe), thereby locating the profiles' troughs and the presumed membrane mid-plane at CLS position Ala311.

The small and polar amino acids, Ser, Thr, and Cys have shallow, nearly neutral profiles, ranging from −0.1 to +0.8 kcal/mol. By contrast, the helix-distorting amino acids, Gly and Pro, which expose the polar protein backbone to the hydrophobic membrane environment, have a high disruptive profile, which peaks (~2 kcal/mol) at the membrane mid-plane, emphasizing the strong unfavorable impact of exposing the polar protein backbone to the membrane environment. The large polar (Asn, His, and Gln) and charged residues (Asp, Glu, Lys, and Arg) are all highly disruptive in the membrane mid-plane. We note that the energetic penalties for Asp, Asn, His, Gln, Glu, and Lys cannot be determined precisely from the dsTβL assay, since the number of reads for substitutions to these residues at the membrane mid-plane in the selected population is nearly 0, reflecting exceedingly large negative-selection pressures (Figure 1; Supplementary file 2).

At the membrane mid-plane, the hydrophobic residues, Val, Ile, Leu, Met, and Phe, show the expected troughs, which are shallower for the small amino acid Val (approximately −0.5 kcal/mol) than for the large amino acids (<−1.5 kcal/mol). We compared the dsTβL values for hydrophobic residues in the membrane mid-plane to values from five hydrophobicity scales (*Figure 2—figure supplement 4*). dsTβL fits well to the Moon scale (*Figure 2c*, $r^2 = 0.90$, with a slope close to 1), which similar to dsTβL measures substitution effects in a bacterial membrane – albeit the outer membrane (*Moon and Fleming, 2011*). The correspondence between dsTβL, which is based on *in vivo* measurement of membrane integration in a bacterial population, with biophysical assays on purified proteins, partly confirms the use of dsTβL for studying membrane-protein energetics.

The dsTβL profile for Trp is similar to the profiles of the aliphatic residues, whereas Tyr makes a nearly neutral contribution to insertion in the membrane core. These profiles diverge from statistical inferences from membrane-protein structures and partitioning experiments, which show that Tyr and Trp preferentially line the membrane-water interface (*Ulmschneider et al., 2005*; *Schramm et al., 2012*; *Senes et al., 2007*; *Nakashima and Nishikawa, 1992*; *Yau et al., 1998*). Further experimental analysis of the role of aromatic residues in membrane-protein stability is warranted, and one possible explanation for these differences is that in the dsTβL assay aromatic residues on the membrane-spanning segment lack neighboring aromatic residues with which to form stabilizing stacking interactions; indeed, experimental stability measurements have shown that stacking makes a significant contribution to the net stabilization provided by aromatic residues in membrane proteins (Hong et al., 2007, 2013).

## Hydrophobicity in the membrane core is similar to that of organic solvents and protein cores

Recently, controversy has surrounded the question of how hydrophobic are biological membranes (*Johansson and Lindahl, 2009*). On the one hand, theoretical considerations and values inferred from hydrocarbons in solution suggested that the free energy contribution due to inserting aliphatic groups into the membrane, or the atomic-solvation parameter, is ~30 ± 5 cal/mol/Å$^2$ of nonpolar surface area (*Vajda et al., 1995*; *Andrew Karplus, 1997*); on the other hand, the Lep measurements suggested values of only 10 cal/mol/Å$^2$ (*Ojemalm et al., 2011*). We analyzed dsTβL data for 39 substitutions from one aliphatic residue (Ala, Val, Ile, Leu, and Met) to another at the

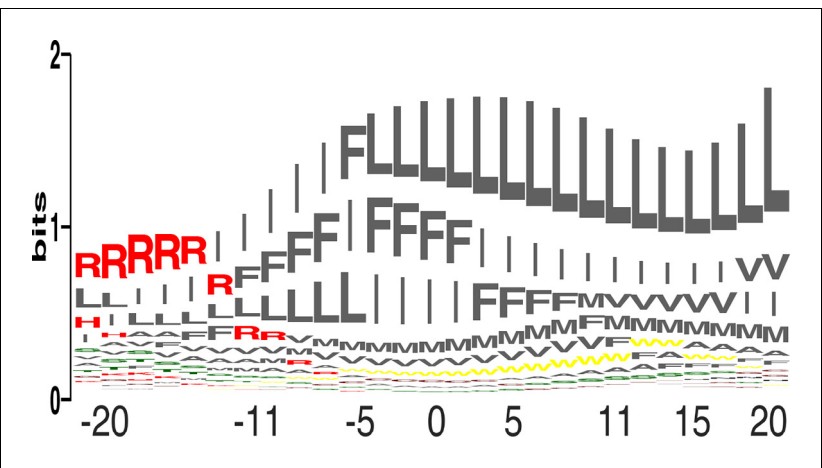

**Figure 3.** Amino acid propensities across the plasma membrane. (a) The relative frequencies of each amino acid within the membrane (see *equation 4* in Materials and methods) in sequence-logo format; the height of each letter corresponds to the amino acid's propensity.

core of the membrane ($-9$ Å$<Z<13$ Å) and inferred an apparent atomic-solvation parameter of $37 \pm 6$ cal/mol/Å$^2$ (*Figure 2d*). We additionally derived an atomic-solvation parameter of $32 \pm 4$ cal/mol/Å$^2$ by analyzing the apparent insertion free energies at the membrane mid-plane ($\Delta\Delta G_{z=0}^{app}$) for each of the aliphatic residues (*Figure 2d*, inset). The values we infer for the atomic-solvation parameter are therefore in fair agreement with values for protein cores and hydrocarbons in aqueous solution (*Vajda et al., 1995*), and 3–4 times larger than the value inferred from the Lep system (*Ojemalm et al., 2011*). We further note that while the ranking of apparent insertion free energies of aliphatic amino acids in dsTβL and Lep (*Hessa et al., 2007*) is similar ($r^2 = 0.79$, *Figure 2—figure supplement 4*), the magnitude of the insertion free-energy changes is nearly four times greater according to dsTβL. We conclude that our results support the view that the hydrophobicity of the plasma membrane core is similar to that of hydrocarbons and much higher than measured in the Lep system.

## Large differences and strong asymmetries in insertion of positively charged residues

A hallmark of plasma membrane proteins is the charge asymmetry known as the 'positive-inside' rule, according to which the cytoplasmic-facing side of the protein is much more positively charged than the periplasmic or extracellular-facing side (*von Heijne, 1989*). This asymmetry has been used to successfully predict the orientation of membrane proteins (*von Heijne, 1989*), but experimental quantification of the energetics of this asymmetry met with difficulty; previous studies measured only a small energy difference (-0.5 kcal/mol) between inserting Arg and Lys in the cytoplasmic relative to the extracellular-facing side of the membrane and no asymmetry for His (*Lerch-Bader et al., 2008*; *Öjemalm et al., 2013*). A striking feature of the dsTβL profiles, by contrast, is that they show clear and large asymmetries for Arg, Lys, and His, in agreement with the 'positive-inside' rule (*Figure 2b*). The three profiles, however, are not identical: whereas Lys and Arg are favored by 2 kcal/mol near the cytoplasm compared to near the periplasm, the titratable amino acid His shows a more modest asymmetry of 1 kcal/mol; moreover, of these three amino acids, only Arg stabilizes the segment near the cytosol, whereas Lys and His are nearly neutral at the cytosol-membrane interface. This difference between Arg and Lys, which has not been noted until now, may be due to charge delocalization in the Arg sidechain and Arg's ability to form more hydrogen bonds with lipid phosphate headgroups.

We compared the relative propensity of each of the 20 amino acids at each position across the membrane (*Figure 3*; *equation 4* in Materials and methods). The results clearly reflect the asymmetric distribution of charges across the plasma membrane, with Arg as the most favored amino acid at the cytoplasmic surface, giving way to the large and hydrophobic amino acids, Leu, Ile, and Phe.

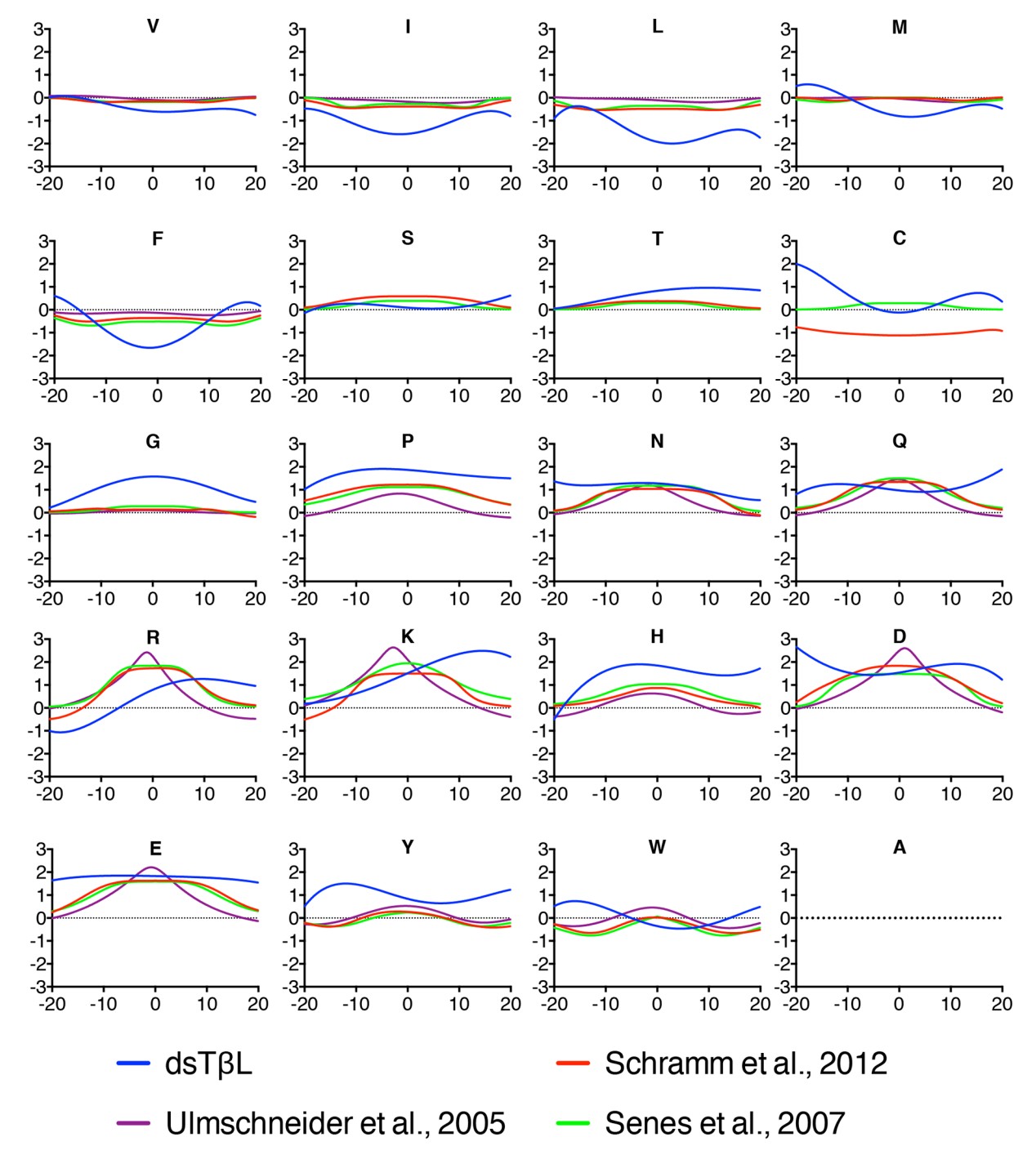

**Figure 4.** Comparison of the dsTβL insertion profiles and knowledge-based profiles of amino acid distributions in membrane-protein structures. Equations for the knowledge-based profiles were taken from *Ulmschneider et al. (2005)*, *Schramm et al. (2012)*, and *Senes et al. (2007)*.

Since the propensities reflect protein-lipid interactions, they could be used to engineer variants of natural membrane proteins that exhibit higher stability and expression levels by mutating membrane-facing positions to the highest-propensity identity. Furthermore, the results suggest that the insertion profiles could be used for bioinformatics prediction of the locations and orientations of membrane-spanning proteins, [manuscript in preparation (Elazar et al.)].

With the accumulation of membrane-protein molecular structures, it has become possible to derive knowledge-based potentials for the insertion of amino acids across the membrane from distributions of amino acids observed in structures (*Ulmschneider et al., 2005*). We compared the dsTβL profiles to three knowledge-based profiles published over the past decade (*Senes et al., 2007*; *Ulmschneider et al., 2005*; *Schramm et al., 2012*) (*Figure 4*). The dsTβL profiles are similar to the knowledge-based ones for the weakly polar and hydrophobic residues (Val, Ser, and Thr), but they diverge with respect to the more hydrophobic and polar residues: for instance, the dsTβL apparent insertion energy for Leu, Ile, and Phe at the membrane mid-plane is −2 kcal/mol and around −0.5kcal/mol for the other scales. Additionally, the only knowledge-based scale that attempted to derive an asymmetric insertion potential (*Schramm et al., 2012*) reported much smaller asymmetries for Lys and Arg, of around 0.5 kcal/mol, compared to 2kcal/mol in dsTβL. These differences are likely

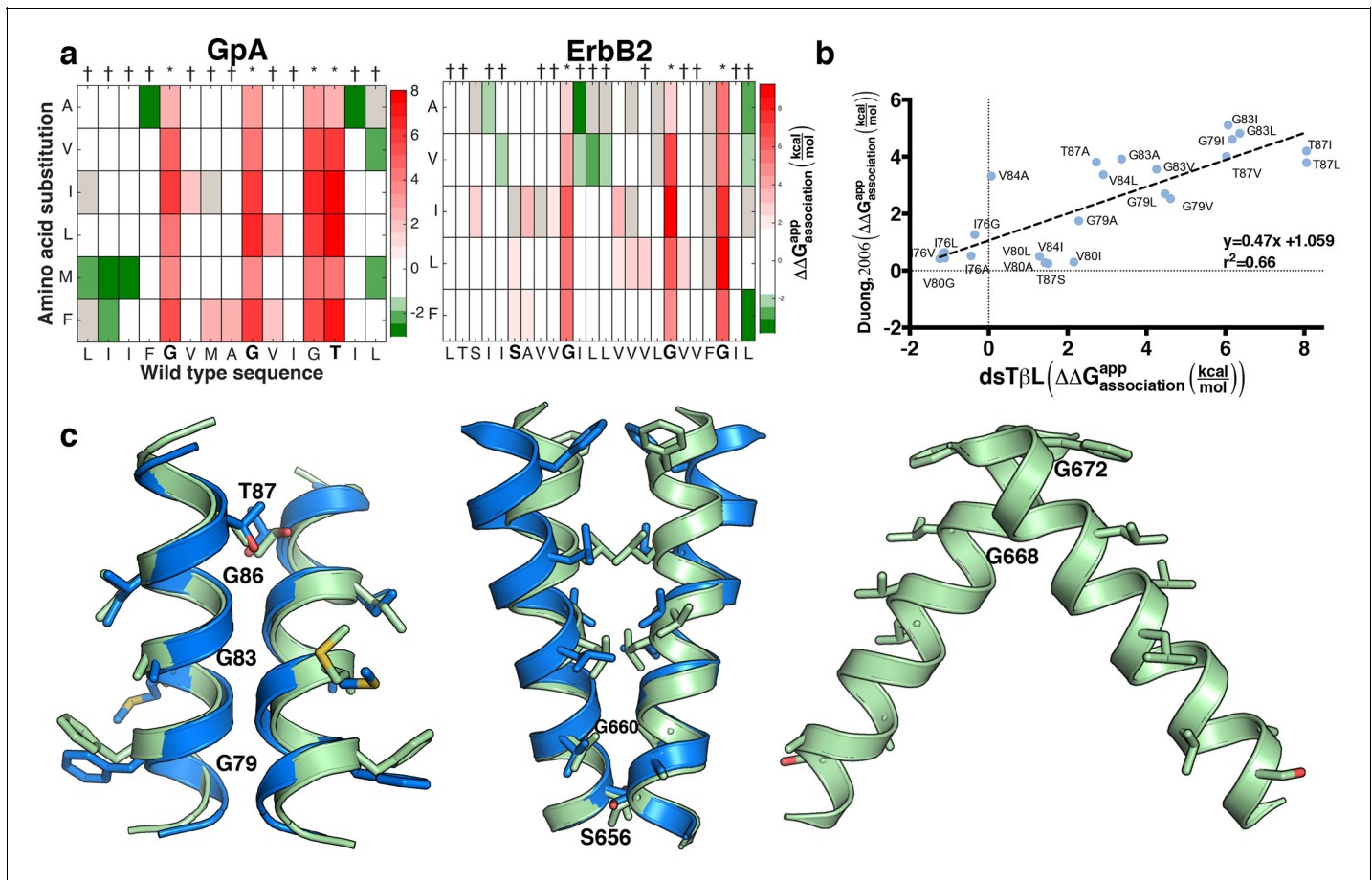

**Figure 5.** Mutational scanning reveals strong coupling between insertion and self-association in receptor homodimers. (a) The mutational landscapes discriminate positions that are involved in self-association (known associating residues are depicted in boldface) from those that do not. (b) Comparison of expression-corrected apparent free energy of insertion for 24 Glycophorin A (GpA) mutants (*Duong et al., 2007*) with results from the dsTβL self-association mutation landscapes. (c) Dimer models that associate through positions that are sensitive to mutation (* in panel a), and do not associate through positions that are insensitive to mutation († in panel a). The models (green) are close to the experimental structures (blue) for Glycophorin A (*MacKenzie et al., 1997*) (1.3 Å root mean square deviation) and ErbB2 (*Bocharov et al., 2008*) (1.9 Å). Another ErbB2 model, which agrees with biochemical and computational evidence (*Endres et al., 2013*; *Arkhipov et al., 2013*; *Fleishman et al., 2002*) but has not been observed in experimental structures, is also suggested.

The following figure supplements are available for figure 5:

**Figure supplement 1.** The mutational landscapes comparing survival on chloramphenicol and ampicillin are dominated by insertion effects.

**Figure supplement 2.** Alternative predicted structures for Glycophorin A and ErbB2.

due to the fact that knowledge-based potentials reflect the frequencies of amino acids in membrane proteins and are biased by functional constraints; indeed polar residues are often found in the membrane core, where they have important roles in oligomerization, substrate binding, transport, and conformational change. Furthermore, the dsTβL experiment is based on a single-pass segment, where every position is exposed to the membrane, whereas the knowledge-based profiles are derived from all structures, including multi-pass proteins, where many residues mediate interactions with other protein segments or line water-filled cavities.

## Strong coupling between insertion and self-association in membrane-spanning homodimers

Insertion and association of membrane-spanning helices are thermodynamically coupled (*Kessel and Ben-Tal, 2002*; *Moll and Thompson, 1994*; *Popot and Engelman, 1990*), but except in one study (*Duong et al., 2007*) these two aspects were assayed separately (*Fleming et al., 1997*; *Finger et al., 2009*; *Hessa et al., 2007*; *Mendrola et al., 2002*). To test the coupling between insertion and association, we applied dsTβL to two model systems for studying membrane protein self-association: the membrane domains of the human erythrocyte sialoglycoprotein Glycophorin A (GpA) and the ErbB2 oncogene, and compared survival on ampicillin and chloramphenicol to survival on ampicillin alone.

Some of the amino acid positions that mediate self-association in GpA and ErbB2 according to their experimentally determined structures (*Bocharov et al., 2008*; *MacKenzie et al., 1997*) show large decreases in chloramphenicol viability upon mutation. Unexpectedly, however, many other mutations that have large effects on chloramphenicol viability are not close to the dimerization surfaces (*Figure 5—figure supplement 1*), suggesting that factors other than self-association dominate the chloramphenicol-viability landscape. To test whether these confounding results are due to variability in expression levels among the mutants, we subtracted from the observed effects of every mutant the expected effects due to expression-level changes according to the dsTβL insertion profiles (see *equation 5* in Materials and methods). The corrected self-association mutational landscapes now correctly discriminate positions that mediate self-association from those that do not (*Figure 5a*), and clearly highlight the interaction motifs in GpA and ErbB2. We compared the results from the systematic self-association landscape of GpA to a previous analysis of 24 GpA mutants that were individually screened for self-association using TOXCAT and corrected for differences in membrane expression using Western blots (*Figure 5b*) (*Duong et al., 2007*). The results from dsTβL and the previous study are consistent ($r^2 = 0.66$; *Figure 5b*), confirming the use of the dsTβL insertion scale to correct self-association mutational landscapes in single-pass homodimers. Furthermore, by systematically probing every position across the membrane, our results highlight additional positions that are sensitive to mutation, such as GpA's Gly86, which was previously not subjected to mutagenesis (*Lemmon et al., 1992*; *Duong et al., 2007*). Moreover, it is notable that although the vast majority of mutations are either neutral or disruptive to self-association, some mutations, for instance to Met in the GpA amino terminus, may promote self-association in the context of the TβL construct (*Figure 5a*). Our results further confirm the strong interplay between membrane-protein expression levels and association, and the importance of accounting for both in biophysical experiments on membrane proteins.

We also tested whether the systematic mutational landscapes generated by dsTβL could be used to provide constraints for structure modeling of receptor membrane domains (*Fleishman et al., 2002*; *Kim et al., 2003*; *Polyansky et al., 2014*). We used the Rosetta biomolecular-modeling software (*Das and Baker, 2008*; *Yarov-Yarovoy et al., 2006*) to generate 100,000 structure models of GpA and ErbB2 directly from their sequences, and selected structures that self-associate through positions that are sensitive to mutation according to dsTβL but not through positions that are insensitive to mutation (*Figure 5c*, *Figure 5—figure supplement 2*). In both cases, fewer than five models passed the selection criteria and of those, some models were within 2 Å of experimentally determined structures.

## Discussion

Despite progress in measuring protein energetics within biological membranes, significant open questions remained, among them, what is the hydrophobicity at the core of biological membranes;

what is the magnitude of the bias for positively charged residues at the cytoplasm surface; and how strong is the coupling between membrane-protein insertion and association energetics? To shed light on these fundamental questions, we established a high-throughput genetic screen and used it to generate systematic mutation landscapes of insertion and self-association in the plasma membrane of live bacteria.

The apparent insertion energies in dsTβL are in line with biophysical stability measurements on outer-membrane proteins (*Moon and Fleming, 2011*), and the inferred atomic-solvation parameter is close to measurements in model systems and protein cores (*Andrew Karplus, 1997*; *Vajda et al., 1995*). Our measurements, however, are three to four times larger than the corresponding ones using the Lep system (*Ojemalm et al., 2011*; *2013*; *Hessa et al., 2007*). To be sure, we are not the first to note these large differences (*Johansson and Lindahl, 2009*; *Shental-Bechor et al., 2006*); yet, we find it significant that our measurements, similar to those in the Lep system, use biological membranes. The observation that the dsTβL insertion measurements for aliphatic side chains have the same ranking but are fourfold larger in magnitude compared to those from Lep (*Figure 2—figure supplement 4*) may indicate that the Lep system measures only a part of the energy contribution to insertion. While further investigation is needed, we speculate that the reason for the large differences between dsTβL and Lep is that total membrane-protein expression levels were not quantified in the Lep system (*Hessa et al., 2007*; *Ojemalm et al., 2011*; *2013*).

We note the following two caveats regarding the dsTβL insertion profiles. First, the penalties for most polar residues at the membrane mid-plane are likely to indicate lower bounds on their insertion energies, since the number of clones counted in the deep-sequencing data for these mutants is close to 0 (supplementary data). Second, statistical analyses (*Ulmschneider et al., 2005*; *Schramm et al., 2012*; *Senes et al., 2007*) and experiments (*Hessa et al., 2007*) demonstrated that the aromatics Tyr and Trp are preferred in the water-membrane interface rather than in the core, although dsTβL shows the reverse (*Figure 2b*). We suggest that these results reflect the fact that dsTβL is based on a monomeric construct where the aromatics are fully exposed to the membrane environment; however, these uncertainties require further research.

The TOXCAT genetic screen has made essential contributions to our understanding of self-association in the membrane (*Lindner and Langosch, 2006*; *Lis and Blumenthal, 2006*; *Russ and Engelman, 1999*; *Finger et al., 2009*; *Mendrola et al., 2002*; *Li et al., 2004*; *Srinivasan et al., 2011 Reuven et al., 2012*). Some early reports demonstrated that chloramphenicol survival also depends on membrane-protein expression levels (*Russ and Engelman, 1999*; *Duong et al., 2007*). Our results strongly support this view and show that expression levels are a dominant factor in chloramphenicol survival. This dominance is perhaps not surprising in retrospect, given that a mutation's effects on monomer concentrations are counted twice in computing its effects on homodimer concentrations, and therefore on chloramphenicol viability (see *equation (5)* in Materials and methods). A key contribution of unbiased and systematic assays, such as dsTβL, is that they clarify such trends unambiguously. Furthermore, the dsTβL insertion profiles derived from the monomeric CLS provide a self-consistent way to factor out the contributions from insertion energetics in future assays on membrane-protein association or function in unrelated membrane proteins, thereby eschewing the need to measure the expression levels of individual mutants.

Deep mutational scanning has made important inroads to analysis and optimization of diverse protein systems (*Whitehead et al., 2012*; *Fowler and Fields, 2014*; *Boucher et al., 2014*). The main strengths of deep mutational scanning are the ability to measure the effects of all point mutations without bias and that all mutants experience strictly equal experimental conditions, thereby limiting experimental noise. The structural simplicity of the model systems tested here, consisting of a single α helix or of helix homodimers, plays a further role in the ability to accurately infer energetics. Combined with structural modeling, the assay can provide essential information both on association energetics and the molecular architecture of membrane receptors. More generally the data on protein-membrane and protein-protein energetics obtained from dsTβL will be used to improve models of membrane-protein energetics and to design, screen, and engineer high-expression mutants of specific membrane proteins (*Fleishman and Baker, 2012*; *Joh et al., 2014*).

# Materials and methods

## Plasmids and bacterial strains

The p-Mal plasmid was generously provided by the Mark Lemmon laboratory. We replaced the maltose-binding protein domain at the open-reading frame carboxy-terminus with β-lactamase (*Lis and Blumenthal, 2006*). The restriction sites in multiple-cloning site 1 were changed to *XhoI* and *SpeI*. The p-Mal plasmid contains a gene for spectinomycin resistance, which is constitutively expressed, providing selection pressure for transformation. The open-reading frame encompassing the TβL construct is also constitutively expressed and is under the control of the weak ToxR promoter.

The DNA coding sequence for the transmembrane constructs used in the paper:

>human CLS

CCGCTGTTCATCCCGGTTGCAGTTATGGTTACCGCTTTTAGTGGATTGGCGTTTATCATCTGGCTGGCT (amino acid sequence: PLFIPVAVMVTAFSGLAFIIWLA)

>Glycophorin A

CTCATTATTTTTGGGGTGATGGCTGGTGTTATTGGAACGATCCTGATC (amino acid sequence: L-IIFGVMAGVIGTILI)

>ErbB2

CTGACGTCTATCATCTCTGCGGTGGTTGGCATTCTGCTGGTCGTGGTCTTGGGCGTGGTCTTTG-GCATCCTGATC (amino acid sequence: LTSIISAVVGILLVVVLGVVFGILI)

The CLS construct was deposited in the *AddGene* repository [pMAL_dstβL-(Plasmid #73805)].

All experiments were conducted using the high-transformation efficiency E. cloni cells (Lucigen Corporation, Middleton, WI).

## Library construction

Customized MatLab 8.0 (MathWorks, Nattick, Massachusetts) scripts for generating primers were written (supplementary files) to generate forward and reverse DNA oligos of lengths 40–85 base pairs, where the central codon is replaced by the degenerate codon NNS, where N is any of the four nucleotides (ATGC) and S is G or C, encoding all possible natural amino acids. Resulting primers were ordered from Sigma (Sigma-Aldrich, Rehovot, Israel). For example, to replace the central 302nd codon of human CLS with an NNS codon, the following two primers were ordered:

>forward

GCTGTTCATCCCGGTTGCAGTT**NNS**TGGTTACCGCTTTTAGTGGATTG

>reverse

CAATCCACTAAAAGCGGTAACCA**SNN**AACTGCAACCGGGATGAACAGC

Each pair of oligos was then cloned into the wild type by restriction-free (RF) cloning (*van den Ent and Löwe, 2006*).

## Transformation, growth, plating, and harvesting

The resulting plasmids from the library-construction step above were electroporated into E. cloni and plated on agar plates containing 50 μg/ml spectinomycin. Plasmids for each position were transformed and plated separately and positions with fewer than 200 colonies were retransformed. All positions were then pooled and used to inoculate 10 ml of Luria Broth medium (LB) with 50 μg/ml spectinomycin and grown in a shaker at 200 rpm and 37°C over-night, diluted 1:1000 and grown to OD = 0.2–0.4. The libraries were then diluted to OD = 0.1 and 200 μl of the resulting cultures were plated at different dilutions (1:1, 1:10, 1:100, 1:1000) on large 12-cm petri dishes containing spectinomycin, ampicillin alone, or ampicillin and chloramphenicol. After overnight incubation at 37°C, p-Mal plasmids were extracted from the resulting colonies using a miniprep kit (Qiagen, Valencia, California).

## Determining concentrations of antibiotics that result in maximal dynamic range

Every wild-type membrane-spanning segment exhibits different sensitivity to chloramphenicol and ampicillin. To determine the concentrations that are most likely to provide maximal dynamic range, we started by cloning mutants that are predicted to reduce insertion of the membrane-spanning segment or its self association (*Mendrola et al., 2002*). Results are represented in

*Supplementary file 1*. We next titrated the wild-type construct as well as the mutant on plates with varying concentrations of antibiotic to find the concentration that shows the largest difference in viability between the wild type and the compromising mutants. *Supplementary file 2* provides the ampicillin and chloramphenicol concentrations used in each of the experiments reported in the paper.

## Deep sequencing

### DNA preparation

In order to connect the adaptors for deep sequencing, the membrane-spanning segments were amplified from the p-Mal plasmids using KAPA Hifi DNA-polymerase (Kapa Biosystems, London, England) using a two-step PCR.

PCR 1:
>forward
CTCTTTCCCTACACGACGCTCTTCCGATCTCTTGGGGAATCGACTCGAG
>reverse
CTGGAGTTCAGACGTGTGCTCTTCCGATCTGTTTAAAGCTGGATTGGCTTGG
1μl of the PCR product was taken to the next PCR step:
>forward
AATGATACGGCGACCACCGAGATCTACACTCTTTCCCTACACGACGC
>reverse barcode 1
CAAGCAGAAGACGGCATACGAGAT <**barcode**>GTGACTGGAGTTCAGACGTGTGC
>reverse barcode 2
CAAGCAGAAGACGGCATACGAGAT <**barcode**>GTGACTGGAGTTCAGACGTGTGC

The DNA samples from each of the populations (unselected; ampicillin-selected; and chloramphenicol and ampicillin selected) were PCR-amplified using DNA barcodes for deep sequencing. The following barcodes were used:

>barcode1
TCGCCAGA
>barcode2
CGAGTTAG
>barcode3
ACATCCTT
>barcode4
GACTATTG

All the primers were ordered as PAGE-purified oligos. The concentration of the PCR product was verified using Qu-bit assay (Life Technologies, Grand Island, New York).

### Deep-sequencing runs

DNA samples were run on an Illumina MiSeq using 150-bp paired-end kits. The quality control for a typical run showed that the membrane-spanning segment was at high-quality (source data) FASTQ sequence files were obtained for each run and customized MatLab 8.0 scripts were written to generate the selection heat maps from the data (scripts are available in supplementary files). Briefly, the script starts by translating the DNA sequence to amino acid sequence; it then eliminates sequences that harbor more than one amino acid mutation relative to wild type; counts each variant in each population; and eliminates variants with fewer than 100 counts in the reference population (to reduce statistical uncertainty). In a typical experiment, at least 70% of the reads passed these quality-control measures.

### Completeness and dynamic range of the deep sequencing results on insertion energetics

The ampicillin selected and the unselected populations of CLS mutants were subjected to deep sequencing analysis yielding more than 4 million reads for each population. Out of 540 possible single-point substitutions, 472 (~87%) mutants were each counted more than 100 times in the reference population; the remaining mutants were eliminated from analysis to reduce uncertainty (gray tiles in *Figure 2a*). The dsTβL assay has a large dynamic range; for instance, at position 307 in the

membrane center, the number of reads in the selected population for Lys, Gln, and Glu is 0, whereas the number of reads for Leu is nearly 110,000, spanning five orders of magnitude.

## Sequencing analysis

To derive the mutational landscapes (*Figures 2a* and *4a*) we compute the frequency $p^{i,j}$ of each mutant relative to wild-type in the selected and reference pools, where *i* is the position and *j* is the substitution, relative to wild-type:

$$p^{i,j} = \frac{\text{count}^{i,j}}{\text{count}_{\text{wild-type}}} \qquad (1)$$

where count is the number of reads for each mutant. The selection coefficients are then computed as the ratio

$$s^{i,j} = \frac{(p^{i,j})_{\text{selected}}}{(p^{i,j})_{\text{reference}}} \qquad (2)$$

where *selected* refers to the selected population (ampicillin in the case of the CLS insertion analysis, and ampicillin plus chloramphenicol in self-association analyses) and *reference* refers to the reference population (spectinomycin-selection in the case of CLS insertion analysis, and ampicillin in the case of self-association analysis). The resulting $s^{i,j}$ values are then transformed to apparent changes in free energy ($\Delta\Delta G^{app}$) due to each single-point substitution through the Gibbs free-energy equation:

$$\Delta\Delta G^{\text{app}}_{i,j} = -\text{RT}\ln(s^{i,j}) \qquad (3)$$

where R is the gas constant, *T* is the absolute temperature (310K), and *ln* is the natural logarithm.

## Polynomial fitting and smoothing of insertion plots

The readout from the insertion selection in dsTβL comprises contributions from the local environment of each position; for example, substitution to a small residue might form a cavity if surrounded by large residues. To reduce such sequence-specific effects, the insertion free-energy values relative to alanine were smoothed using the MatLab smooth function over a window of 5 residues (2 on each side), excluding gray tiles (with insufficient data), and plotted as points in *Figure 2b*. The points were then fitted using the polynomial fitting function polyfit to yield 4th-order polynomials (*Figure 2b* lines and *Supplementary file 1*). Two centrally located polar amino acid positions, CLS positions Ser307 and Gly308, were discarded from the analysis due to their inconsistency with the general trends of the insertion profiles, likely because mutations at these polar positions distort the helix backbone.

## Position-specific amino acids preference

To compute the amino acid preference at each position in the membrane (*Figure 2c*), we calculated the Boltzmann-weighted probability of every amino acid residue at each position in the membrane-spanning domain of human CLS(*Srinivasan et al., 2011*) using the following formula (MatLab script in supplement files):

$$p^{i,j} = \frac{e^{-E^{i,j}_{\text{app}}/\text{RT}}}{\sum_x e^{-E^{i,x}_{\text{app}}/\text{RT}}} \qquad (4)$$

where R is the gas constant, *T* = 310K and $E^{i,j}_{app}$ are the apparent free energy of transfer of amino acid *j* at position *i* relative to alanine (*Figure 2b*).

## Inferring the atomic-solvation parameter

We generated a model of the CLS membrane domain by threading its sequence on a canonical α helix, and used Rosetta to singly introduce each substitution from one aliphatic identity (Ala, Val, Ile, Met, Leu, and Phe) to another in the membrane core. Amino acid sidechains were combinatorially repacked and the change in solvent-accessible surface area (ΔSASA) was computed. Four additional data points (marked with asterisks, *Figure 2d*) were extracted from Glycophorin A's position Ala82,

which is located at the membrane center and away from the dimerization interface. To compute the atomic-solvation parameter from the insertion energies of the aliphatics at the membrane mid plane (*Figure 2d*, inset), we compared the insertion energy at the membrane mid plane for each aliphatic residue $\Delta\Delta G_{z=0}^{app}$ with computed $\Delta$SASA of a change from that residue to Ala on a canonical poly-Ala α helix.

## Computing the apparent dimerization free-energy change

ToxR activity depends on homodimer concentrations (*Langosch et al., 1996*; *Russ and Engelman, 1999*), and homodimer concentrations depend on both monomer insertion into the membrane and self-association strength. The measured effects of every point mutation on self-association (*Figure 5—figure supplement 1*) therefore comprise contributions from insertion (multiplied by two because the homodimer comprises two mutants) and dimerization (*Duong et al., 2007*). To isolate the mutation's effects on self-association (*Figure 5a*), we subtract from every data point twice the contribution to insertion at the relevant position along the membrane normal.

$$\Delta\Delta G_{\text{dimerization}_{j,x}}^{\text{app}} = \Delta\Delta G_{\text{measured}_{j,x}}^{\text{app}} - 2\Delta\Delta G_{\text{insertion}_{i,j,x}}^{\text{app}} \tag{5}$$

where, $i$, $j$, and $x$ are the wild-type identity, mutation, and the position along the membrane normal, respectively, $\Delta\Delta G_{measured}$ is the measured change in self-association free energy (see *equation (3)*), and $\Delta\Delta G_{insertion}$ is the free-energy change expected for a mutation from $i$ to $j$ at position $x$ according to the insertion polynomials of *Supplementary file 1*.

## β –lactamase blot analysis

Cells were grown in 5 ml LB overnight at 37°C. The cells were then diluted at a 1:100 ratio into 50 ml LB and were grown to $A_{600}$ = 0.6, harvested on ice, washed in TBS buffer, and equal amounts of cells were re-suspended in extraction buffer (50 mM Tris pH 8 [Bio-Lab, Israel], 100 mM NaCl, 5% [w/w] sucrose and 1 mM AEBSF [Sigma-Aldrich]). The cells were then disrupted by three cycles of sonication with Microson XL at 12 watts for 10 s with 60 s intervals. Samples were centrifuged for 15 min at 13,000 rpm in order to discard cell debris, supernatant was ultracentrifuged for 1 hr at 300,000 g using Optima TLX with TLA100.1 rotor in order to sediment membranes to the pellet. The pellet was re-suspended in 100 mM Na(HCO$_3$)$_2$ and incubated for 15 min at 4°C and ultra-centrifuged for 1 hr at 300,000 g. Pellet and extract protein concentration were measured with Lowry protein assay (*Peterson, 1977*), and equal amounts of protein were loaded on 12.5% Tris-Glycine SDS PAGE gels. Gels were transferred to Protran nitrocellulose membranes (Whatman) and incubated with mouse anti-β-lactamase antibody (Santa Cruz, Dallas) and a horse-radish-peroxidase-fused rabbit anti-mouse secondary antibody, and imaged using SuperSignal West Femto Maximum Sensitivity Substrate (Thermo, Waltham, MA). ECL and the chemiluminescence signal was detected using the ChemiDoc MP System (Bio-Rad). Band densitometry was analyzed with imageJ (*Schneider et al., 2012*).

## *Ab initio* modeling using Rosetta

For each membrane segment, we start by generating all-helical backbone-conformation fragments using the Rosetta utility fragment picker (*Gront et al., 2011*) and construct a C2 symmetry definition file using the Rosetta symmetry utility function (https://www.rosettacommons.org). We then use the Rosetta Fold-and-Dock application (*Das et al., 2009*), which samples symmetric degrees of freedom for both docking and folding of the homodimer using the RosettaMembrane energy function (*Yarov-Yarovoy et al., 2006*). Example files and command lines for running fragment picker and fold-and-dock are available in supplementary files.

## Constraining structure models with experimental results

For each position in the membrane-spanning region of the target protein, we assign two labels: *likely mediating binding* – if at least four substitutions from wild-type disrupted binding by at least 2kcal/mol (* in *Figure 5a*); and *unlikely to mediate binding* – if at least four substitutions improved or did not change binding (†). For each of the 20% lowest-energy Rosetta models, we tested whether at least two of the residues that likely mediate binding are within 5 Å of the partner monomer and all positions, which are unlikely to mediate binding, are outside a 4 Å shell. Structures that

passed the filter above were clustered using the Rosetta clustering application with default parameters. The clusters were visually inspected and models showing significant kinks were eliminated.

## Acknowledgements

We thank Olga Khersonsky, Dror Baran, Adina Weinberger, Jens Meiler, and Zohar Mukamel for advice, and Gunnar von Heijne, Nir Ben-Tal, Ingemar Andre, Julia Koehler-Leman, and Dan Tawfik for critical reading. Ilan Samish and William DeGrado provided advice for analyzing knowledge-based insertion energies. The research was supported by the Minerva Foundation with funding from the Federal German Ministry for Education and Research, a European Research Council's Starter's Grant, an individual grant from the Israel Science Foundation (ISF), the ISF's Center for Research Excellence in Structural Cell Biology, career development awards from the Human Frontier Science Program and the Marie Curie Reintegration Grant, an Alon Fellowship, and a charitable donation from Sam Switzer and family.

## Additional information

### Funding

| Funder | Author |
|--------|--------|
| Minerva Foundation | Assaf Elazar<br>Jonathan Weinstein<br>Yearit Fridman<br>Sarel Jacob Fleishman |
| European Research Council | Assaf Elazar<br>Jonathan Weinstein<br>Yearit Fridman<br>Sarel Jacob Fleishman |

The funders had no role in study design, data collection and interpretation, or the decision to submit the work for publication.

### Author contributions

AE, Conducted the insertion and association deep sequencing experiments, Conducted the western blot experiments, Wrote software to analyze the deep-sequencing data, Conception and design, Acquisition of data, Analysis and interpretation of data, Drafting or revising the article; JW, Analyzed structure-based potentials, Acquisition of data, Analysis and interpretation of data, Drafting or revising the article; IB, Conducted the western blot experiments, Acquisition of data, Analysis and interpretation of data, Drafting or revising the article; YF, Conception and design, Acquisition of data; EB, Conception and design, Analysis and interpretation of data, Drafting or revising the article; SJF, Conception and design, Acquisition of data, Analysis and interpretation of data, Drafting or revising the article

### Author ORCIDs

Sarel Jacob Fleishman, http://orcid.org/0000-0002-6831-3770

## Additional files

### Supplementary files

• Supplementary file 1. Polynomial fit for the insertion profiles of the 20 amino acids. The insertion data (*Figure 2b*, points) were fitted to the following fourth-order polynomial, where $Z$ is given in Å ($Z = 0$ at the membrane midplane) and $\Delta G$ is in kcal/mol: $\Delta G_{insertion}^{app} = a_0 Z^4 + a_1 Z^3 + a_2 Z^2 + a_3 Z + c$ Parameter $c$ is the value of insertion at the membrane midplane. The right-hand column reports the $r^2$ value for the polynomial fit to the data points. The fit for Glu (E) is poor, reflecting the flatness of the Glu profile (*Figure 2b*).

• Supplementary file 2. Wild-type and mutants used in dsTβL experiments. For each membrane-spanning wild-type segment we optimized the selection stringency. Mutated sequences were used as negative controls in experiments to identify optimal selection regimes, where the difference in bacterial growth between wild-type sequence and mutant was largest.

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
