## [Decision Letter]

Thank you for submitting your work entitled "Mutational scanning reveals the determinants of protein insertion and association energetics in the plasma membrane" for consideration by *eLife*. Your article has been reviewed by two peer reviewers, and the evaluation has been overseen by Yibing Shan (Reviewing Editor) and John Kuriyan as the Senior Editor. One of the two reviewers, William DeGrado, has agreed to share his identity.

The reviewers have discussed the reviews with one another and the Reviewing editor has drafted this decision to help you prepare a revised submission.

Summary:

This paper describes a mutational scanning analysis to determine the energetics of membrane insertion of transmembrane helices and helix-helix association in membrane. Much of the method is new, which uses a modified TOXCAT assay coupled with high throughput screening and deep sequencing. The results suggest a membrane hydrophobicity higher than previously estimated, energetic asymmetry for several amino acids between the periplasmic and the cytoplasmic lipid layers, and a significant difference in the membrane energetics of Arg and Lys. The helix-helix association data confirm that the GxxxG motif represents favored dimerization interfaces. These results may inform analysis and design of membrane proteins and they are of broad interest to the community.

Essential revisions:

The reviewers suggested a number of important revisions that would help strengthen this manuscript.

1) The reviewers pointed out an important caveat for the present analysis: strictly speaking, the results should be considered propensities rather than free energies of membrane insertion and helix-helix association, without calibrating the expression level, or the relative population of the membrane-inserted protein vs the protein in cytosol. We suggest the authors Western Blot the expression levels of a small set (~10) of mutants and hopefully to show small variances in the expression levels. In any event this caveat should be discussed in the revised manuscript.

2) Both reviewers thought the discussion concerning TM topology is a distraction from the main focus. It would be better to remove this discussion from this manuscript and to publish it separately with further development.

3) The results reported in Figure 2 (Z-dependent energetic profiles) should be analyzed in comparison with bioinformatics results (Senes et al., 2007; Ulmschneider et al. 2005; Ulmschneider et al. 2006; Schramm et al., 2012).

4) The reviewers asked for further description of the present dsTβL system and for a more detailed comparison of the present system with the previously developed Lep system. Why the choice of β-lactamase instead of the maltose binding protein? What is the precise sequence of L-selectin used in the system? Why the high sensitivity of membrane insertion to even conservative TM mutations? The authors attribute the lower membrane hydrophobicity estimated by previous work (Hessa et al., 2007) to translocon interactions of the Lep system. Please elaborate this point, as the present system also passes through translocons to be embedded in the membrane.

Individual reviewer comments:

Reviewer #1:

Sarel Fleishman et al. present a mutational scanning analysis to determine the energetics of TM helix insertion into a membrane. Using a modified TOXCAT assay coupled with high throughput screening and deep sequencing they have derived insertion profiles for each amino acid at all positions within a selected TM helix. Based on these results they conclude that (i.) membranes are more hydrophobic than anticipated based on previous findings, and (ii.) Arg and Lys differ in their membrane insertion asymmetry. Furthermore, the authors have identified an asymmetry for several amino acids and have used their data to predict the membrane topology of a set of *E. coli* proteins with experimentally determined TM topologies. In addition, the authors have used their insertion energetics together with association energetic to model the structure of two selected TM helix dimers.

I found the first part, i.e. the selection of amino acids and the determination of membrane insertion preferences, rather convincing, albeit some open questions remain, which are outlined in detail below. However, the authors mention throughout the entire manuscript many times that they present "the first systematic mutation analysis of insertion and self-association in the plasma membrane of live bacteria". This is simply wrong, and the entire idea is not that new. ToxR-based assays are in use for quite a while and it is common that authors present some analysis of the proteins expression levels. Thus, the entire statement, that membrane insertion and association energetics have never been studied in parallel simply is wrong. In fact, in MT Doung et al. (JMB (2007) 371, 422) the authors have adjusted their measured CAT activities to the expression levels. Furthermore, in Finger et al. (JMB (2006) 358, 1221) the authors have used a system very similar to TOXCAT and they have systematically followed the changes associated with changing expression levels. Thus, I do not see the new concept here. Similarly, I found the results of the modeling little convincing (subsection “High-precision structure modeling constrained by systematic self-association measurements”, last paragraph). Especially for small TM helix dimers there are many good predictions out, and I do not see that the current modeling results are superior.

I feel that the authors should consider concentrating on the first part of the manuscript and on the prediction of the TM topology. In the subsection “Topology prediction” they mention that another manuscript is in preparation. They might consider combining these two manuscripts, which would certainly improve the current manuscript. The analysis of the TM topology would certainly enhance the visibility of the present work.

Open questions:

1) In the second paragraph of the Introduction and later on: It is mentioned several times that the differences observed between the current results and previous results (Hessa et al. 2007) are due to interactions between the engineered protein and other components of the membrane, such as the translocon. However, in the present study the authors also express a fusion protein and a large part of that has to pass the translocon. While the fusion proteins analyzed in the present study certainly differ from the Lep systems, as we have a soluble domain, one could argue that the passage of the large soluble protein domain into the periplasm somehow influences the results.

Along the same lines: I do not understand the argumentation that the lower membrane hydrophobicity see with the Lep system is due to the translocon (Discussion, second paragraph). Also in *E. coli* the TM segments are inserted into the membrane via the SecYEG translocon! Please clarify.

2) In the first paragraph of the subsection “dsTβL: a high-throughput assay for measuring membrane-protein energetics”: I agree that the orientation of the segment within the membrane is known due to the system set up. However, the question arises whether the authors have missed all the proteins in their analysis which have a different topology. Thus, does the entire assay really determine insertion energetic or energetics of insertion in only one defined way?

3) System set up: When the equilibrium between soluble and membrane inserted residue is somehow affected by an amino acid, thus when a residue forces (or abolishes) insertion of a TM helix into the membrane, some β-lactamase might still be expressed into the periplasm (just less or more). As far as I understand the assay, one would still select the clones, as they grow on the selective agar, and add their sequence to the analysis. How meaningful is an analysis of residues, which are only partly integrated into the membrane are also expressed as soluble versions? Why should there be a negative selection if the selection pressure allows growth of all constructs? This problem might also influence the later presented interaction analysis. In principle the authors mention and use this argument in a different context in the Discussion (second paragraph).

4) In the subsection “High-precision structure modeling constrained by systematic self-association measurements”, second paragraph: It is clear that some mutations severely affect the expression level of a given protein. While this is here presented in the context of dimerization, a reduced expression level might also affect survival on ampicillin plates. So the question arises, what is actually measured and compared? Membrane insertion propensity of an amino acid or the impact of a substitution on the expression level? Probably both.

5) When "mutations form polar residues distort the helix backbone" and these are therefore discarded, how valuable is then the analysis of any polar residue in the present analysis. Is the argument valid for all positions?

*Reviewer #2:* This paper describes an elegant approach to screen for helix-helix interactions in the membrane, which relies on deep-sequencing of populations that have been selected for transmembrane peptide insertion and/or dimerization as well as the unselected controls. Much of the method is new. The results are interesting and can help enable the design and analysis membrane proteins; they should be of broad interest to the community. However, I have many suggestions to improve the clarity of the presentation, the interpretation of the results, and some of the computational methods briefly presented in the paper. Although these comments are lengthy, I do feel this is a very interesting and important paper that should be rapidly published.

The authors used an unusual membrane protein in this study to compare and contrast with earlier study based on Lep. The system is based on the single-span membrane protein, ToxR, which lacks a signal sequence. The construct contains a water-soluble N-terminal DNA-binding domain, a transmembrane helix, and a C-terminal β-lactamase (bla) as a periplasmic selectable marker. Thus, the entire system is in reverse order from the more common Type I insertion process, with no efficient signal protein to direct insertion into the membrane. The protein system studied here is also different from other bacterial Type II membrane proteins in that it has a full N-terminal domain. One strong point of this paper is that it allows comparison of this unusual insertion process with previous systems. However, the authors do not discuss these differences, and there is no discussion of whether their system is cotranslationally or post-translationally inserted, whether it requires SecA and SecB, and whether it uses the SecYEG.

The authors use an elegant series of selections to define the background population, the pool of protein sequences that insert properly, and the pool of protein sequences that both insert and dimerize. The authors point out that much of this has been accomplished previously using other methods, but not in a high-throughput manner. (Also the Bouchner manuscript was cited in such a way that it seemed that some of the present work was presented previously, which can easily be fixed.) The authors do not comment on why MBP is removed from the starting construct and β lactamase is used instead. Did they find growth +/- maltose to be an insufficient screen?

The constructed system would appear, perhaps serendipitously, to be very sensitive to small changes. Even an Ala to Leu change in the TM sequence can have a large change selective advantage, which almost certainly relates to the efficiency of insertion. Nevertheless, I am not sure whether they have shown this using a membrane prep. and Western analysis of a dozen variants spanning different insertion efficiencies, which would be standard in most papers on this subject. Also, the authors do not address whether the L-selectin's TM sequence is acting as a cleaved signal peptide or whether it is retained as a TM tether for the bla domain.

The extreme sensitivity to single mutations through the entire TM sequence is unusual, as similar studies generally show only localized regions to be important. Indeed, if any conservative mutation in a TM sequence led to a significant change in fitness it would be difficult to understand how proteins evolve and show such sequence variability. I therefore became curious what the actual sequence for L-selectin's TM was used. Following the gene sequence from the Methods section,

AGCCAGCCAGATGATAAACGCCAATCCACTAAAAGCGGTAACCATAACTGCAACCGGGATGAACAGCGG

and translating in all three possible reading frames, the most hydrophobic peptide sequence would be:

ASQMINANPLKAVTITATGMNS

There are many polar amino acids in this stretch. This clearly is not the sequence that was used by Renhao Li's lab in the reference provided in the manuscript [1]. Either there is a major problem in what was intended versus what was actually cloned, or this is a typo that could have led to confusion in the literature. Either way, the correct amino acid sequence should be shown, along with the adjacent sequence N-terminal and C-terminal to the insert. If the native TM from human L-selectin was indeed cloned properly, the authors might wish to discuss how such small changes make large differences in fitness. Normally, one would expect that the sequence would need to be teetering on the edge of stability or function to be so sensitive throughout its sequence.

The authors would like to use the counts of amino acid types at a given position in the sequence observed in the control population versus that in the bla-selected pool to compute the free energy of insertion of the TM helix. There is good reason to believe that this reflects the insertion efficiency (although even this is difficult to be sure of without Western analysis for confirmation). However, it is a real stretch to equate this with true free energies. For insertion efficiency to be interpreted as free energies then the system would need to be under thermodynamic or pseudo-thermodynamic equilibrium. This was achieved in the Lep system by quantifying the two populations of inserted states, and in Karen Fleming's system by quantifying the concentration of the inserted and un-inserted populations of protein. In this case, however, we have no idea the concentration of the un-inserted protein and whether the rate-determining step in this unusual case represents engagement with accessory proteins, insertion into the translocon, or release from the translocon. Furthermore, the concentration of soluble protein in the cell vs. the protein inserted into the membrane is not measured. I feel, the authors should report their primary data as propensities or frequencies and restrict their conclusions to statements like: "if we treat these insertion frequency signatures as if they were under thermodynamic control following a Boltzmann we find that…" Plots derived based on this assumption are fine, but only after appropriate discussion of the caveats.

The authors do not compare their Z-dependent position profiles for the individual amino acids to the corresponding frequency plots obtained from bioinformatics analysis of membrane protein crystal structures [2-5]. Also, it is not clear how they decide on the Z-position in the membrane for their inserted TM helix. Clearly it depends on the surrounding sequence, which was not provided. Comparing the present scale to bioinformatics scales, the distributions behave in the expected manner for hydrophobic residues, which is reassuring. Also, if the propensities from earlier papers are converted to energies using the somewhat questionable method of reverse Boltzmann statistics employed in the present paper (see supplement of [5]) they would appear to support the magnitude of the solvation parameter being similar to that found in Karen Fleming's analysis, supporting the analysis presented in the current paper. This is a striking finding, and one of the reasons I feel that the current paper should be accepted with revision. However, beyond this, the differences between the present scale and previous experimental or bioinformatics scales are not subtle. Tyr and Trp do not show more favorable interaction in the headgroup region than the center of the bilayer in the current analysis, even though it is seen in virtually every other experimental and bioinformatics study. Glu, Asp, Gln, Asn are essentially flat, and do not become more favorable as the depth of insertion approaches +/- 20 Å as in previous studies. If this is due to averaging and is not significant, the profiles should not be shown. The profiles for Lys and Arg are also different from expectation. ΔΔG for Arg vs. Ala is around 0 at -5 Å (very close to the center of the bilayer!) and Lys is unfavorable in the cytoplasmic headgroup region. This must be telling us something about the specific system they are studying, which should be expanded on. I very much doubt that these data can be used alone for making general conclusions, although when appropriately analyzed to understand differences with other systems they will be quite informative.

Figure 2 shows a nice correlation with the Moon Fleming values, but it is not clear what is plotted for dsTBL (ΔΔG at Z=0?). Figure 2 would appear to contain a small selection of the data from a very large number of potential sequences that have been obtained from the deep sequencing. What is the slope and correlation if all sequences are used, or if it is computationally difficult how do they change after choosing 25 sequences versus 50 versus 100 sequences? What if the smoothed value of DDG at Z=0 is used to compute the values? What if one plots the X-values of Figure 2 vs. the area of the sidechains in a poly-Ala helical backbone?

Do the data in Figure 4 teach us anything new that would not have been gleaned from Mark Lemmon's, Don Engelman's and Axel Brunger's first papers in the early 1990s or Kevin MacKenzie's and Karen Fleming's subsequent papers [6-11] or does this serve as a calibration exercise? How do the data in Figure 4 compare to the analysis of Mark Lemmon [12] and subsequent papers?

High-precision structure modeling constrained by systematic self-associationmeasurements. In reading this section one is struck by the fact that the computational power required is many orders of magnitude greater than what was used by Axel Brunger to analyze glycophorin A two decades ago [13, 14], and yet the derived structures are not as close in RMSD to those predicted by his group prior to the structure determination of glycophorin. His method also works on asymmetric sequences. Also, the authors do not discuss methods that use mutagenesis results in a prospective way to predict structures, which, again, appear to perform better than the current method [15-17].

"Since dsTβL is the first experimental scale to report large insertion asymmetries for positively charged residues we examined whether the profiles could be used to predict membrane-protein topology directly from sequence using a benchmark comprising 607 bacterial membrane proteins of experimentally determined topology (Daley et al. 2005). The dsTβL-based predictor (see Methods) correctly assigns topology in ~70% of the cases, within the performance range of statistics-based predictors (70-80%) (Tsirigos et al. 2015). Prediction accuracy increases to 81% where the computed energy gap between inserting the protein in one orientation or the other is larger than 3kcal/mol (Figure 3). It is encouraging that the dsTβL predictor, which is based on an experimental scale measured on a single-pass membrane protein, performs on par with methods that were fitted using experimental topology data. We are currently testing the ability of dsTβL to predict other structural properties of membrane proteins directly from sequence."

These prediction values are poorer than what is seen using analogous structural bioinformatics scales (e.g., 5), and similar to what is seen when using hydrophobicity alone. If the differences between Lys and Arg are indeed generally applicable one would expect an improvement. In any event, this one subsection appears to be the topic of a future paper (stated by the authors in the Methods section). I am sure there will be sufficient space to properly compare their novel prediction method to existing methods in this future paper. As is, I think this small subsection detracts from what is otherwise an interesting paper.

In conclusion, this is a very interesting paper describing the first (?) application of deep-sequencing methods to the problem of helix-helix interactions in the membrane. Given the novelty and importance of the work, I have endeavored to provide some perspective and suggestions in this review. I would urge the editors to publish the paper. Even if published as is, I don't think it would be a fine addition to the literature, and the issues I address would ultimately be worked out in future papers.

Finally, the authors should comment on the large difference in the length of the inserted sequences used as TM helices for glycophorin A versus ErbB2.

Glycophorin: LIIFGVMAGVIGTILI (16 residues)

ErbB2: LTSIISAVVGILLVVVLGVVFGILI (25-residues)

References:

[1] Srinivasan, S., Deng, W., and Li, R. (2011) L-selectin transmembrane and cytoplasmic domains are monomeric in membranes, Biochim Biophys Acta 1808, 1709-1715.

[2] Senes, A., Chadi, D. C., Law, P. B., Walters, R. F., Nanda, V., and Degrado, W. F. (2007) E(z), a depth-dependent potential for assessing the energies of insertion of amino acid side-chains into membranes: derivation and applications to determining the orientation of transmembrane and interfacial helices, J Mol Biol 366, 436-448.

[3] Ulmschneider, M. B., Sansom, M. S., and Di Nola, A. (2005) Properties of integral membrane protein structures: derivation of an implicit membrane potential, Proteins 59, 252-265.

[4] Ulmschneider, M. B., Sansom, M. S., and Di Nola, A. (2006) Evaluating tilt angles of membrane-associated helices: comparison of computational and NMR techniques, Biophys J 90, 1650-1660.

[5] Schramm, C. A., Hannigan, B. T., Donald, J. E., Keasar, C., Saven, J. G., Degrado, W. F., and Samish, I. (2012) Knowledge-based potential for positioning membrane-associated structures and assessing residue-specific energetic contributions, Structure 20, 924-935.

[6] Lemmon, M. A., Flanagan, J. M., Treutlein, H. R., Zhang, J., and Engelman, D. M. (1992) Sequence Specificity in the Dimerization of Transmembrane α Helices, Biochemistry 31, 12719-12725.

[7] Treutlein, H. R., Lemmon, M. A., Engelman, D. M., and Brunger, A. t. (1992) The glycophorin A transmembrane domain dimer: sequence-specific propensity for a right-handed supercoil of helices, Biochemistry 31, 12726-12733.

[8] Lemmon, M. A., and Engelman, D. M. (1994) Specificity and promiscuity in membrane helix interactions, Quarterly reviews of biophysics 27, 157-218.

[9] Lemmon, M. A., Treutlein, H. R., Adams, P. D., Brunger, A. T., and Engelman, D. M. (1994) A dimerization motif for transmembrane α helices, Nature, Structural biology 1, 157-163.

[10] Mingarro, I., Whitley, P., Lemmon, M. A., and von Heijne, G. (1996) Ala-insertion scanning mutagenesis of the glycophorin A transmembrane helix: a rapid way to map helix-helix interactions in integral membrane proteins, Protein Sci 5, 1339-1341.

[11] MacKenzie, K. R., and Fleming, K. G. (2008) Association energetics of membrane spanning α-helices, Curr Opin Struct Biol 18, 412-419.

[12] Mendrola, J. M., Berger, M. B., King, M. C., and Lemmon, M. A. (2002) The single transmembrane domains of ErbB receptors self-associate in cell membranes, J Biol Chem 277, 4704-4712.

[13] Adams, P. D., Arkin, I. T., Engelman, D. M., and Brunger, A. T. (1995) Computational searching and mutagenesis suggest a structure for the pentameric transmembrane domain of phospholamban, Nature Structural Biology 2, 154-162.

[14] Adams, P. D., Engelman, D. M., and Brünger, A. T. (1996) Improved prediction for the structure of the dimeric transmembrane domain of glycophorin obtained through global searching, Proteins 26, 257-261.

[15] Berger, B. W., Kulp, D. W., Span, L. M., DeGrado, J. L., Billings, P. C., Senes, A., Bennett, J. S., and DeGrado, W. F. (2010) Consensus motif for integrin transmembrane helix association, Proc Natl Acad Sci U S A 107, 703-708.

[16] Metcalf, D. G., Law, P. B., and DeGrado, W. F. (2007) Mutagenesis data in the automated prediction of transmembrane helix dimers, Proteins 67, 375-384.

[17] Soto, C. S., Hannigan, B. T., and DeGrado, W. F. (2011) A photon-free approach to transmembrane protein structure determination, J Mol Biol 414, 596-610.

---

## [Author Response]

*The reviewers suggested a number of important revisions that would help strengthen this manuscript. 1) The reviewers pointed out an important caveat for the present analysis: strictly speaking, the results should be considered propensities rather than free energies of membrane insertion and helix-helix association, without calibrating the expression level, or the relative population of the membrane-inserted protein vs the protein in cytosol. We suggest the authors Western Blot the expression levels of a small set (~10) of mutants and hopefully to show small variances in the expression levels. In any event this caveat should be discussed in the revised manuscript.*

We agree with this caveat and have addressed it in several ways:

A) We adopted the argument recommended by Reviewer #2 that the free energy derivation depends on the constancy of the cytosolic amounts of protein among the different mutants (subsection “dsTβL: a high-throughput assay for measuring membrane-protein energetics”, last paragraph) and we explicitly mention now the assumption of thermodynamic control of partitioning between the different states. We further state that this assumption is supported by the agreement between the results and biophysical measurements.

B) We verified the behavior of 10 selected mutants by Western blots of membrane preparations (Figure 2—figure supplement 3). The Western blots showed that the proteins express in the membrane, and all run at the expected size, thereby excluding the possibility of cleavage. 6 mutations that target membrane-core positions and one at the amino terminus show the expected trends, including mutations that increase or decrease expression. However, 3 mutants to charges at the amino terminus increased expression levels according to Western blots, but were disruptive according to dsTβL. We propose in the revised manuscript that these results are due to the fact that ampicillin selections probe appropriate integration in the membrane, not just expression levels as in Western blots.

C) The reviewers found it surprising that the screen is so sensitive, even to mild mutations. In the first paragraph of the subsection “Systematic per-position contributions to membrane-protein insertion” we now note that the CLS amino acid sequence is quite polar, suggesting that its membrane-expression levels would be sensitive even to point mutations. To further confirm this sensitivity, for the 10 above-mentioned mutants we carried out plate-viability assays on a clone-by-clone basis, and in 9 saw the same behavior as in the deep sequencing experiments (Figure 2—figure supplement 2), including complete non-viability for some of the mutants.

*2) Both reviewers thought the discussion concerning TM topology is a distraction from the main focus. It would be better to remove this discussion from this manuscript and to publish it separately with further development.*

Right. We replaced this discussion with the statement: “…the results suggest that the insertion profiles could be used for sequence-based prediction of the locations and orientations of membrane-spanning proteins (A.E., J.W, et al., manuscript in preparation).”

*3) The results reported in Figure 2 (Z-dependent energetic profiles) should be analyzed in comparison with bioinformatics results (Senes* et al.*, 2007; Ulmschneider* et al. *2005; Ulmschneider et al. 2006; Schramm* et al.,

*2012).*

This now appears in Figure 4 and discussed in the last paragraph of the subsection “Large differences and strong asymmetries in insertion of positively charged residues”. Briefly, we note some similarities but also significant differences with the dsTβL profiles. Most importantly the statistics-based profiles are quite flat by comparison and show minor contributions for hydrophobicity or the positive-inside rule. We discuss these differences with respect to the fact that membrane-protein statistics reflect functional constraints rather than pure energetics. Reviewer #2 noted that our profiles for Tyr and Trp did not match the expectation that these residues are favored at the membrane-water interfaces. We agree with this point and discuss it in the last paragraph of the subsection “Systematic per-position contributions to membrane-protein insertion”, suggesting that these observations may reflect differences between single and multi-span membrane proteins. We also summarized caveats regarding the insertion scales in the Discussion, third paragraph, which we hope will clarify these points and spur future research.

*4) The reviewers asked for further description of the present dsTβL system and for a more detailed comparison of the present system with the previously developed Lep system. Why the choice of* β

*-lactamase instead of the maltose binding protein? What is the precise sequence of L-selectin used in the system? Why the high sensitivity of membrane insertion to even conservative TM mutations? The authors attribute the lower membrane hydrophobicity estimated by previous work (Hessa et al., 2007) to translocon interactions of the Lep system. Please elaborate this point, as the present system also passes through translocons to be embedded in the membrane.*

A) We have elaborated our description of the TβL construct (subsection “dsTβL: a high-throughput assay for measuring membrane-protein energetics”). Briefly, there were two reasons for using β-lactamase instead of MBP: 1. Viability on ampicillin is linearly related to β-lactamase expression levels whereas MBP on maltose is not (see refs. provided in the paper); and 2. The TOXCAT-MBP construct requires working with an MBP-null *E. coli* strain (MM39), which is not amenable to high-throughput genetic transformations, whereas with β-lactamase we could work with the E. cloni high-transformation efficiency strain.

B) As mentioned above in our response 1C, we verified the high sensitivity on a clone-by-clone basis. The reason for this high sensitivity is that the CLS amino acid sequence is quite polar.

C) Regarding our comparison to the Lep system: we have revised our argument in the Discussion. Briefly, the Lep measurements quantified proteins that were either singly or doubly glycosylated, but did not quantify total membrane-protein levels. The observations that the Lep measurements are rank-ordered as in our measurements (and also in published biophysical experiments on an outer-membrane protein), but are fourfold smaller in magnitude, as well as the observation that the atomic-solvation parameter inferred from Lep is smaller by roughly fourfold compared to both dsTβL and previous biophysical work are consistent with a view that Lep was measuring only part of the insertion equilibrium. We have removed the argument that the differences are due to interactions with the translocon.

D) Reviewer #2 was correct in identifying a mistake in the DNA sequence we provided for CLS in the original supplement, which was of the CLS reverse-complement. We corrected this mistake and apologize for the error. The amino acid sequence of CLS is noted at the bottom of Figure 2.

In addition to the changes above, we corrected, clarified, and reanalyzed our data following the individual reviewer comments, including those that were not mentioned in the summary. With respect to the self-association data we now compare to the Doung et al. data on GpA (Figure 5) finding high correspondence between their data and ours. We also revised this section to highlight the new findings in dsTβL, including positions, which were not tested in the Lemmon et al. and Doung et al. studies, but are sensitive to mutation, or the few mutations that promote self-association; without a comprehensive analysis it is unlikely to identify the very few beneficial mutations. We further shifted the emphasis from structure prediction to the insertion-association coupling by shortening the structure-prediction paragraph and changing the last sentence in the Abstract and the sub-title of the relevant section. We feel that the self-association section is an important and integral part of this paper because it shows that a single assay can easily provide systematic data on the two primary components of membrane-protein energetics: insertion and association. Furthermore, the fact that we used the insertion data derived from CLS to subtract the insertion contribution in two unrelated systems (GpA and ErbB2) confirms the general usefulness of the insertion scales and should simplify future studies that analyze mutational effects on association or function.